# Effect of a Multi-Strain Probiotic on the Incidence and Severity of Necrotizing Enterocolitis and Feeding Intolerances in Preterm Neonates

**DOI:** 10.3390/nu14163305

**Published:** 2022-08-12

**Authors:** Marwyn Sowden, Mirjam Maria van Weissenbruch, Andre Nyandwe Hamama Bulabula, Lizelle van Wyk, Jos Twisk, Evette van Niekerk

**Affiliations:** 1Department of Global Health, Division of Human Nutrition, Faculty of Medicine and Health Sciences, Stellenbosch University, Cape Town 7505, South Africa; 2Amsterdam UMC, Department of Pediatrics-Neonatology, VU University Medical Center, 1081 HV Amsterdam, The Netherlands; 3Infection Control Africa Network—ICAN, Cape Town 7530, South Africa; 4Department of Paediatrics and Child Health, Stellenbosch University, Cape Town 7505, South Africa; 5Department of Epidemiology and Data Science, Amsterdam UMC, 1081 HV Amsterdam, The Netherlands

**Keywords:** necrotizing enterocolitis, feeding intolerance, neonate, probiotic

## Abstract

Background: Necrotizing enterocolitis (NEC) is a multifactorial disease, causing inflammation of the bowel. The exact root of NEC is still unknown, but a low weight and gestational age at birth are known causes. Furthermore, antibiotic use and abnormal bacterial colonization of the premature gut are possible causes. Premature neonates often experience feeding intolerances that disrupts the nutritional intake, leading to poor growth and neurodevelopmental impairment. Methods: We conducted a double-blind, placebo-controlled, randomized clinical trial to investigate the effect of a multi-strain probiotic formulation (Labinic^TM^) on the incidence and severity of NEC and feeding intolerances in preterm neonates. Results: There were five neonates in the placebo group who developed NEC (Stage 1A–3B), compared to no neonates in the probiotic group. Further, the use of probiotics showed a statistically significant reduction in the development of feeding intolerances, *p* < 0.001. Conclusion: A multi-strain probiotic is a safe and cost-effective way of preventing NEC and feeding intolerances in premature neonates.

## 1. Introduction

Necrotizing enterocolitis (NEC) is an inflammatory bowel disease, often seen in the neonatal intensive care unit (NICU). The majority of NEC cases (>90%) occurs in premature neonates with a birthweight and gestational age below 1500 g and 32 weeks. The reported global incidence of NEC in premature neonates with a gestational age below 32 weeks is in the range 2–7.6% [1,2,3]. The reported NEC incidence in our unit, in 2021, was 6.3%.

NEC is the leading cause of mortality due to gastrointestinal disease in the NICU [4]. Owing to its multifactorial nature, the exact etiology of NEC is still unknown [5,6]. The most important factor associated with NEC is prematurity. Other postnatal factors include the following: immunodeficiency, antibiotic use, dysbiosis, choice of feed as well as delayed enteral feeding, low birth weight <1500 g, caesarean section birth, neonatal stress, factors involving intestinal ischemia/hypoxia and inflammation [5,7,8,9,10]. These factors lead to a disruption of the intestinal mucosal integrity and cause intestinal ischemia that is clinically manifest as described in Bells criteria [9,11,12,13]. Desfrere et al., showed in a preliminary report that premature neonates born to human immunodeficiency virus (HIV)-positive mothers have an increased risk of developing NEC [14].

The pathogenesis of NEC can also strongly be linked to abnormal bacterial colonization—a reduced gut diversity and altered bacterial strains, predominantly with pathologic bacteria [7,8,15]. The preterm neonates’ initial exposure to environmental microorganisms is most often from the clinical surroundings (equipment, air, neonates, and nursing personnel acting as carriers) [16], leading to an altered intestinal microbiota with significantly less *Bifidobacteria* present [17]. An increase in *Proteobacteria*, especially *Enterobacteriaceae* species, preceding the diagnosis of NEC, have been found [15,18]. Steward et al. confirmed that antibiotics had a substantial effect on microbiome changes, i.e., reduced *Escherichia species* and increased *Enterobacteriaceae species* [15].

A high percentage of premature neonate’s experience feeding intolerances clinically evident by abdominal distention, or emesis, or both and a disruption of the neonate’s feeding plan [19,20].

The use of probiotics for various gastrointestinal diseases, e.g., Crohn’s disease, ulcerative colitis, irritable bowel disease, infectious diarrhea, *Clostridium difficile* infection, antibiotic associated diarrhea, NEC, pouchitis associated with inflammatory bowel disease, infantile colic, etc. looks promising. However, it is important to note that not all probiotics are beneficial in all circumstances, as different strains have different biological effects. One thus needs to select specific organism(s) as an appropriate treatment for different disease prevention or treatment [8,9,10,13,21,22,23].

Numerous systematic reviews and meta-analyses have reported that probiotics can significantly reduce NEC Stage 2 and above, as well as feeding intolerance. However, probiotics are not used as standard of care despite its cost effectiveness in reducing morbidity, mortality, and long-term complications such as poor growth and neurodevelopmental impairment [24].

The aim of this study was to investigate the effect and clinical relevance of the supplementation of a multi-strain probiotic (Labinic^TM^) consisting of *Lactobacillus acidophilus* (0.67 billion colony forming units (CFU)s), *Bifidobacterium bifidum* (0.67 CFUs) and *Bifidobacterium infantis* (0.67 CFUs) on the incidence and severity of NEC and feeding intolerances in preterm neonates in a low-resource healthcare setting, such as South Africa. We focused on neonates <1500 g, since data on this age group is limited.

## 2. Materials and Methods

### 2.1. Study Design

A double-blind, placebo-controlled, randomized clinical trial. The primary aim of the study was to determine the impact of a multi-strain probiotic administration on rectal colonization with drug-resistant Gram-negative bacteria in preterm neonates. One of the secondary aims of the study is described in this article, namely, to determine the effect of a multi strain probiotic in reducing the incidence and severity of feeding intolerances and NEC in preterm neonates.

### 2.2. Study Setting

The study was conducted at Tygerberg Hospital (TBH). TBH can host 1384 patients, of which 132 beds are for neonates. The neonatal wards include a 12-bed medical/surgical NICUs, 2 high-care wards, 1 low-care ward and 1 kangaroo mother care (KMC) ward. Participants were recruited from the two high care wards and data was collected over a period of 6 months, from 19 January to 27 June 2021.

### 2.3. Study Participants

Male and female preterm neonates, with a birth weight between 750–1500 g and <37 weeks gestation was enrolled in the study. Neonates with major congenital malformations, early onset sepsis (C-reactive protein (CRP) >10 mg/L in the first 72 h of life) [25], preterm neonates up for adoption, neonates with major gastro-intestinal abnormalities or requiring surgery of the gastro-intestinal tract were not eligible for inclusion.

### 2.4. Randomization

Neonates were assigned to the probiotic (intervention) (*n* = 100) or placebo group (*n* = 100) using random number allocation (Figure 1). The researcher and all neonatology staff were blinded to the group assignments. Consecutive sampling was used, i.e., every preterm neonate meeting the inclusion criteria described above was selected until the required sample size groups was achieved.

### 2.5. Procedures

The probiotic used was Labinic^TM^ (Biofloratech, Surrey, UK) and the placebo consisted of medium chain triglyceride (MCT) oil and Aerosil 200 (Aerosil 2000 is a stabilizer used in Labinic^TM^ as well). Labinic^TM^ consists of *Lactobacillus acidophilus* (0.67 billion colony forming units (CFU)s), *Bifidobacterium bifidum* (0.67 CFUs) and *Bifidobacterium infantis* (0.67 CFUs). The manufacturer ensured similarity of the probiotic and placebo packaging to ensure blinding of clinical personnel. Unblinding of package labelling occurred after conclusion of the study.

At the time of the study, the standard of care was no probiotic administration. The standard dose of 0.2 ml of probiotic was administered once daily for 28 days, providing 2 billion colony forming units per day. Supplementation with the probiotic or placebo was delayed if the neonate was nil per os (NPO) and discontinued if a neonate developed NEC (Bell’s stage II or more). The probiotic/placebo was added to the neonate’s feed (mother’s own breast milk, donor breast milk or infant formula) and administered via an orogastric tube or if applicable, orally. Neonates were followed up from birth to a maximum of 28 days, death, discharge to peripheral hospitals or home, whichever time point came first.

Data collected included neonatal demographic data, feeding data (type and volume), anthropometric data, medication, clinical and laboratory data. All data were collected daily. As per neonatal protocol, nasogastric residuals are not measured. For the purpose of this study, feeding intolerance was identified when abdominal distension and/or emesis was encountered and lead to a disruption of the feeding plan. The color, volume, and frequency of vomits and open nasogastric drainage was noted, as well as abdominal distension, stool volume and consistency [26].

Neonates were evaluated as medically indicated for the development of NEC by the attending neonatologists. If the diagnosis of NEC was made, it was staged according to Bell’s classification [11]. The study neonatologist served as consultant at ward level for any clarification.

Under the feeding regime, the day that feeds were started was described as the day of first feed received, and the first day that the neonate received 160 mL/kg or more was described as full feeds reached. In TBH hospital, it is not practice giving standard total parenteral nutrition (TPN) to all very low birth weight (VLBW) neonates, and the hospital’s practice is early and aggressive enteral feeds with mothers’ own (expressed breastmilk or if applicable pasteurized expressed breastmilk) or donor breast milk and formula feeds only when there is no alternative after day 5. TPN is reserved for surgical neonates only. 

### 2.6. Statistical Analysis

The current study was a secondary outcome of the main study (currently in review, not yet published) aiming to determine a reduction in carriage rate of antibiotic-resistant Gram-negative organisms with the supplementation of a multi-strain probiotic. The total sample size of the main study was 200, with 100 neonates per group (treatment and placebo groups). It was estimated using a published decrease (17%) in the proportion of rectal colonization with drug-resistant bacteria [27]. This sample size was estimated to detect a significant difference between the groups being compared (with Type I error at 0.05 and power at 80%). The total sample size required allowed a 12% margin for study participant lost to follow-up.

An intention to treat analysis comparing probiotic vs. placebo treated was performed for all outcomes. For NEC and feeding intolerance development, logistic regression analysis was used. Abdominal distention days as well as vomit days showed a discrete and highly skewed-to-the-right distribution; thus, for these outcomes, negative binomial regression analysis was used. For all outcomes, both crude and adjusted analyses were performed. Adjustments were made for sex, HIV exposure, birthweight, delivery method, gestational age, day of start of feeds, day of start of KMC and feeding type. For all statistical tests performed, a *p*-value < 0.05 was considered significant. For descriptive analyses, we summarize baseline characteristic variables using mean (SD) or median (IQR) as appropriate (based on the normality of data distribution), as well as proportions for categorical variables. All the statistical analyses were performed using STATA 16.0 (Statacorp, College Station, TX 77845, USA).

## 3. Results

The demographics of the study cohort are shown in Table 1. There were no differences in demographics between the probiotic and placebo groups. The mean gestational age between the two groups was similar: probiotic group, 29 weeks, ±13.9 days (range 25–36 weeks); placebo group, 30 weeks, ± 13.5 days (range 25–34 weeks). The mean birthweight was also similar between the two groups: probiotic, 1174 ± 226 g (range 780–1500 g); placebo, 1150 ± 230 g (range 750–1495 g). The majority of neonates were born via a caesarean section (*n* = 146; 73%). The percentage of neonates who received empiric antibiotic therapy at birth for possible infection, was similar between the two groups (placebo, *n* = 54 (54%) versus probiotic, *n* = 57 (57%)). Table 1 describes the basic demographic information of the 200 neonates enrolled in the study.

Few (n = 5) neonates developed NEC (Table 2) and no statistical analysis was therefore performed. However, none of the neonates developed NEC in the probiotic group during the 28-day study period. In contrast, five neonates in the placebo group developed NEC: stage 1A: two neonates; stage 1B, stage 2A and stage 3B: one neonate each. One neonate in the placebo group passed away due to stage 3B NEC with multi-organ failure. Table 3 described the characteristics of the neonates that developed NEC, Bell’s stage I to III. None of the neonates that was HIV exposed or had a gestational age > 33 weeks developed NEC.

Although the mean day of feed initiation was slightly later in the probiotic group, these neonates reached full feeds earlier than the placebo group (Table 4). The probiotic group was less likely to be placed NPO or to receive TPN.

A lower number of neonates in the probiotic group was diagnosed as having a feeding intolerance (Table 5). Abdominal distention observed in the probiotic group was halved and less emesis was observed in the probiotic group.

The results of the analyses show that there was a statistically significant difference in the development of feeding intolerances, number of days as well as number of emesis observed, days with abdominal distention as well as number of days NPO between the two groups. Number of days on TPN was not statistically significantly different. The use of probiotics showed a statistically significant reduction in the development of feeding intolerances (both crude and adjusted); *p* < 0.001. Gender, delivery method, days when feeds or KMC was started and feeding type showed no significant influence. Birth after 33 weeks was a protective factor against feeding intolerances in these neonates. Neonates born before 33 weeks showed a higher incidence of feeding intolerances odds ratio (OR): 0.34 (0.15 to 0.82); *p* = 0.015. Neonates born with a higher birthweight (above 1000 g) were less likely to present with feeding intolerances OR: 0.37 (0.14 to 0.94); *p* = 0.036. A total of 1871 probiotic doses were administered (mean doses 18.7 per neonate) during the study period of 28 days (minimum of 4 and maximum of 28 doses).

No protocol changes, violations nor serious adverse events relating to the use of the probiotic occurred.

## 4. Discussion

In our study, the incidence of NEC (Bells staging II or more) in the probiotic group was zero cases, while the incidence in the placebo group was two cases, of which one neonate passed away due to NEC complications. There was thus a reduction in NEC cases (Bells staging II or more) from 6.3% reported in 2021 to 2% in the placebo and 0% in the probiotic group in our study. The overall reduction might be due to cross-colonization in the placebo group [28]. A recent systematic review shows that around 7% of preterm neonates develop NEC worldwide [29].

An increase in the prevalence of NEC has been previously described in HIV-exposed neonates, especially in the presence of antiretroviral therapy [14,30]. Of interest to note is that none of the HIV-exposed neonates on antiretroviral therapy in our study developed NEC. A study investigating the effect of probiotics on NEC in HIV-exposed neonates concluded that probiotics reduced the severity of NEC in the HIV-exposed study group, while reducing the incidence of NEC in both HIV-exposed and -unexposed premature VLBW neonates [31]. This agrees with our findings that a multi-strain probiotic reduced NEC in both HIV-exposed and -unexposed premature VLBW neonates.

A systematic review and meta-analysis by Aceti et al. showed varying effects between probiotic trials and the prevention of NEC, and no definitive conclusion could be drawn [8]. However, other systematic review and meta-analysis studies showed positive effects on the use of certain probiotic strains. AlFaleh et al. showed that the administration of *Lactobacillus* species alone as well as a mixture of probiotic strains significantly reduced the incidence of severe stage II to III NEC. The administration of a mixture of probiotics reduced the incidence of mortality; however, the administration of *Lactobacillus* or *Bifidobacterium* species alone did not reduce mortality [7]. Bi et al. showed that *Lactobacillus* reduces the incidence of NEC but advised that *Bifidobacterium* probiotic mixture might be the preferred option for NEC, sepsis, and all-cause death reduction in premature neonates [32]. Sawh et al. also concluded that NEC reduction was seen in neonates receiving *Lactobacillus*, *Bifidobacterium* species or multispecies supplements [33].

One study in the United Kingdom, by Robertson et al. showed in a single-center retrospective observational study that the use of a triple-species probiotic consisting of *L. acidophilus, B. bifidum, and B. longum subspecies infantis* can lead to a 4.4% reduction in NEC [34]. Another study in Bolombia by Hoyos used Infloran (*Bifidobacteria infantis* and *Lactobacillus acidophilus*). They showed a reduction in NEC from 6.6% (85 NEC cases out of 1282 participants in the control group) to 3.0% (34 NEC cases out of 1237 participants in the probiotic group) [35].

The use of a multi-strain probiotic might thus be a viable answer to successfully decrease NEC in this vulnerable population.

The establishment of enteral feeding in premature neonates are often delayed [36]. 

Our study showed that the use of a multi-strain probiotic effectively shortens the time to reach full feeds, reduces the development of feeding intolerances, number of days as well as number of emesis observed, days with abdominal distention as well as number of days NPO compared to a placebo. Totsu et al. also showed in a multicenter clinical trial that the use of *Bifidobacterium bifidum* accelerates the establishment of enteral feeding after birth, without increasing adverse effects [36]. A systematic review and meta-analysis by Athalye-Jape et al. indicated that 19 out of 25 trials concluded that the use of probiotics can reduce the time to full enteral feeds, fewer episodes of feeding intolerances, improved weight gain and growth velocity, decreased transition time from orogastric to breast feeds, increased postprandial mesenteric flow and a reduced hospital stay [24]. Neonates supplemented with probiotics (irrespective of *Bifidobacterium* or non-*Bifidobacterium* strains; single or multiple strains or early and late initiation of probiotics) took less time to achieve full feeds compared to the placebo groups (mean difference of 1.54 d) [24]. Methods by which probiotics can decrease feeding intolerances include increased gut maturity and gut motility by increased intestinal transit time and increased gastric emptying as well as increased superior mesenteric artery flow [24].

In conclusion, our study showed that Labinic^TM^ can effectively be used as a cost-effective intervention in reducing NEC and feeding intolerances in neonates <1500 g.

A limitation of this study that needs to be acknowledged is the high proportion of the study population that were transferred out to peripheral hospitals, owing to high occupancy rates at the tertiary hospital, which led to reduced days of observation during the trial. Study strengths included that the probiotic/placebo administration was blinded to all involved and we managed to include VLBW and extremely low birth weight (ELBW) neonates in the study. A suggestion for future research is to include microbiome analysis, in order to assess and confirm colonization with the probiotic administered and to assess cross-colonization in the placebo group.

## 5. Conclusions

Our study showed that the use of a multi-strain probiotic effectively shortens the time to reach full feeds, reduces the development of feeding intolerances, number of days as well as number of emesis observed, days with abdominal distention, number of days NPO and incidence of NEC compared to a placebo.

The full trial protocol is available from the main author.

## Figures and Tables

**Figure 1 nutrients-14-03305-f001:**
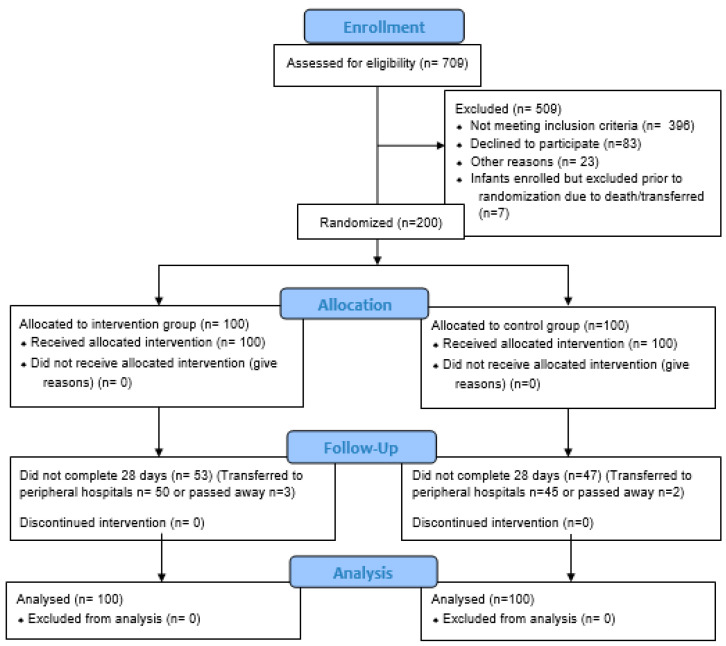
Consort diagram shows the screening process of participants, through enrollment up to analysis.

**Table 1 nutrients-14-03305-t001:** Basic demographic information.

	Probiotic Group (*n* = 100)	Placebo Group (*n* = 100)
Gender		
Male (*n*, %)	47 (47)	37 (37)
Birth weight		
750–1000 g (*n*, %)	30 (30)	32 (32)
1001–1500 g (*n*, %)	70 (70)	68 (68)
Gestational Age		
26–28 weeks (*n*, %)	34 (34)	30 (30)
29–32 weeks (*n*, %)	60 (60)	62 (62)
33–36 weeks (*n*, %)	6 (6)	8 (8)
Apgar score (10 min)		
<4 (*n*, %)	0 (0)	1 (1)
4–7 (*n*, %)	10 (10)	9 (9)
>7 (*n*, %)	89 (89)	89 (89)
No Apgar (born before arrival (*n*, %)	1 (1)	1 (1)
HIV		
Exposed (*n*, %)	22 (22)	26 (26)
Birth number		
Single neonate (*n*, %)	79 (79)	86 (86)
Twin neonates (*n*, %)	21 (21)	14 (14)
Reason for premature delivery		
SPPROM (*n*, %)	16 (16)	20 (20)
FD (*n*, %)	57 (57)	43 (43)
EOPET (*n*, %)	2 (2)	4 (4)
Placenta abruption (*n*, %)	2 (2)	7 (7)
IUGR (*n*, %)	1 (1)	6 (6)
SPTL (*n*, %)	18 (18)	18 (18)
HELLP (*n*, %)	2 (2)	1 (1)
Placenta praevia (*n*, %)	2 (2)	1 (1)
Empiric antibiotic use for presumed sepsis at birth		
Neonates classified as at septic risk at birth and received empiric antibiotics (*n*, %)	57 (57)	55 (55)
Days (mean days ± SD)	3.8 ± 2.1, (range 1–12)	3.8 ± 2.0, (range 1–10)

EOPET: Early onset pre-eclampsia; FD: Fetal distress; HELLP: Hemolysis, elevated liver enzymes, low platelet count; IUGR: Intrauterine growth restriction; SD: Standard deviation; SPPROM: Spontaneous preterm premature rupture of the membranes; SPTL: Spontaneous preterm labor.

**Table 2 nutrients-14-03305-t002:** NEC complications in the neonates.

	Probiotic Group (*n* = 100) *n* (%)	Placebo Group (*n* = 100) *n* (%)
NEC	0 (0%)	5 (5%)
NEC Bell’s diagnosed Stage 1A	0	2 (2%)
NEC Bell’s diagnosed Stage 1B	0	1 (1%)
NEC Bell’s diagnosed Stage 2A	0	1 (1%)
NEC Bell’s diagnosed Stage 3B	0	1 (1%)
Neonate passed away due to NEC	0	1 (1%)

**Table 3 nutrients-14-03305-t003:** Characteristics of the neonates with NEC Bell’s stage I, II and III.

	Bell’s I	Bell’s II	Bell’s III
Treatment group			
Probiotic	0	0	0
Placebo	3	1	1
HIV-exposure			
HIV-exposed	0	0	0
HIV-unexposed	3	1	1
Gestational age			
26–28 weeks	1	0	1
29–32 weeks	2	1	0
33–36 weeks	0	0	0
Birth weight (grams)			
750–1000 g	2	0	1
1001–1500 g	1	1	0

**Table 4 nutrients-14-03305-t004:** Description of feeding regime of the neonates.

	Probiotic Group	Placebo Group
	(*n* = 100)	(*n* = 100)
Day first feed received (days), (mean days ± SD)	3.1 ± 1.1	3.00 ± 1.0
Day full feeds reached (days), (mean days ± SD)	8.7 ± 2.0	9.7; ± 4.3
Duration of TPN (days) (mean days ± SD)	0.1 ± 1.0	0.5 ± 2.2

**Table 5 nutrients-14-03305-t005:** Feeding intolerance in the probiotic in placebo group.

	Probiotic Group (*n* = 100) *n* (%)	Placebo Group (*n* = 100) *n* (%)
Feeding tolerance	86 (86%)	65 (65%)
Feeding intolerance diagnosed (number of neonates)	14 (14%)	35 (35%)
Emesis observed (number of neonates)	49 (49%)	69 (69%)
Abdominal distension observed (number of neonates)	24 (24%)	48 (48%)
Neonates NPO for 1 or more days due to emesis (number of neonates)	8 (8%)	20 (20%)
Neonate received TPN (number of neonates)	2 (2%)	5 (5%)
Duration NPO (days) (mean ± SD)	0.2 ± 0.4	0.4 ± 0.7
	Crude Odds Ratio	Adjusted Odds Ratio *
Development of feeding intolerance	0.19 (0.09 to 0.43); *p* < 0.001	0.11 (0.04 to 0.30); *p* < 0.001
	Crude Rate Ratio	Adjusted Rate Ratio *
Number of days with emesis	0.41 (0.28 to 0.60); *p* < 0.001	0.39 (0.26 to 0.56); *p* < 0.001
Number of emesis observed	0.40 (0.26 to 0.43); *p* < 0.001	0.35 (0.23 to 0.54); *p* < 0.001
Days with abdominal distention	0.20 (0.11 to 0.35); *p* < 0.001	0.20 (0.12 to 0.35); *p* < 0.001
Number of days NPO	0.35 (0.15 to 0.84); *p* = 0.019	0.27 (0.10 to 0.71); *p* = 0.008

* Adjusted for the following variables: Sex, HIV exposure, birthweight, delivery method, gestational age, day of start of feeds, day of start of KMC and feeding type.

## Data Availability

Data supporting reported results can be requested from the main author.

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
