# Peer review of "Effect of a Multi-Strain Probiotic on the Incidence and Severity of Necrotizing Enterocolitis and Feeding Intolerances in Preterm Neonates"

_nutrients, 2022, doi:10.3390/nu14163305_

Round 1
Reviewer 1 Report
This is an interesting paper with analysis of the intervention to prevent NEC in premature neonates.
Within the placebo arm, the NEC incidence is 5%. If paper goal is to conclude the probiotic impact on NEC incidence, the author should introduce the NEC incidence in the premature neonate population in the introduction. The reported incidence of NEC in preterm infants with a GA <32 weeks varies globally from 2 to 7.5 percent across different neonatal intensive care units (NICUs).
Another issue is the dosing effect. If authors can demonstrate the dosing effect of the probiotics, then the intervention effect would be more conclusive.
Please revise the manuscript to harmonize the above two points.
Reviewer 2 Report
Important study, well designed. Excellent methods and results section, but the introduction needs some improvement and the discussion needs significant improvement.
Specific comments:
Describe antibiotic use between the treatment and control group, assess for any differences or clearly state if there were no antibiotics used. If there are differences in antibiotic use between treatment and control - you will need to reconsider most if not all of your results.
Has the main study been published? If so, cite it. If not, state that.
Introduction: There have been many systematic reviews/meta-analyses in this space. Clearly state what questions this study will answer beyond what is already in the literature and state what this study adds to the literature.
Discussion: The discussion is just a list of results from a literature review. Please see other discussions as examples and rewrite the discussion to reflect the content that should be in a discussion.
Round 2
Reviewer 2 Report
Good response to suggestions, could still use a stronger justification for why this is novel or adds to the literature in the introduction. However, not a deal breaker for me.
Author Response
Thank you for the feedback. The justification was strengthened Introduction line 110-115.
